# Prediction of Prognosis and Chemotherapeutic Sensitivity Based on Cuproptosis-Associated lncRNAs in Cervical Squamous Cell Carcinoma and Endocervical Adenocarcinoma

**DOI:** 10.3390/genes14071381

**Published:** 2023-06-30

**Authors:** Jianghong Zhou, Lili Xu, Hong Zhou, Jingjin Wang, Xiaoliang Xing

**Affiliations:** 1Department of Gynecology, Department of Obsterics and Gynecology, Zhuzhou Hospital Affiliated to Xiangya School of Medicine, Central South University, Zhuzhou 412007, China; zhoujianghong84@163.com (J.Z.); xll1230404@126.com (L.X.); 15576518668@163.com (H.Z.); wjjzzzxyy@163.com (J.W.); 2School of Public Health and Laboratory Medicine, Hunan University of Medicine, Huaihua 418000, China

**Keywords:** CESC, cuproptosis, prognosis, immune, chemotherapy

## Abstract

Cervical cancer is the fourth most common cancer. The 5-year survival rate for metastatic cervical cancer is less than 10%. The survival time of patients with recurrent cervical cancer is approximately 13–17 months. Cuproptosis is a novel type of cell death related to mitochondrial respiration. Accumulative studies showed that long non-coding RNAs (lncRNAs) regulated cervical cancer progression. Compressive bioinformatic analysis showed that nine cuproptosis-related lncRNAs (CRLs), including C002128.2, AC002563.1, AC009237.14, AC048337.1, AC145423.1, AL117336.1, AP001542.3, ATP2A1-AS1, and LINC00426, were independently correlated with the overall survival (OS) of cervical squamous cell carcinoma and endocervical adenocarcinoma (CESC) patients. The time-dependent area under curve value reached 0.716 at 1 year, 0.718 at 3 years, and 0.719 at 5 years. Notably, CESC patients in the low-risk group had increased immune cell infiltration and expression of several immune checkpoints, which indicated that they may benefit more from immune checkpoint blockade therapy. In addition, we also used the model for drug sensitivity analysis. Several drug sensitivities were more sensitive in high-risk patients and showed significant correlations with the risk models, such as Bortezomib_1191, Luminespib_1559, and Rapamycin_1084, suggesting that these drugs may be candidate clinical drugs for patients with a high risk of CESC. In summary, this study further explored the mechanism of CRLs in CESC and provided a more optimized prognostic model and some insights into chemotherapy of CESC.

## 1. Introduction

Cervical cancer remains the fourth most common cancer among women, with 604,000 new cases of cervical cancer and 342,000 deaths reported in 2020 [1,2]. Human papillomavirus is an important factor in cervical cancer [3,4]. The integration of the human papillomavirus genome into host cells leads to abnormal regulation of cell cycle control and can also induce immune escape from infected cells [5]. Persistent HPV infection induces immune tolerance of the host immune system, which is one of the important mechanisms of cervical lesions [5]. Other important co-factors include some sexually transmitted infections (e.g., HIV, chlamydia trachomatis), smoking, having a high number of children, and long-term use of oral contraceptives [1]. In recent years, benefiting from the HPV vaccines, early screening and effective surgery, chemotherapy and radiotherapy, the survival rate of cervical cancer patients has improved significantly [6]. The 5-year survival rate is about 70%. But the 5-year survival rate for metastatic cervical cancer is less than 10% [7]. The survival time of patients with recurrent cervical cancer is approximately 13-17 months [8].

TNM staging plays a crucial role in the clinician’s CESC treatment plan. Unfortunately, due to the high heterogeneity of CESC, cases of the same TNM stage for CESC patients may have very different clinical outcomes [9]. Therefore, there is an urgent need to find suitable accurate biomarkers for the early detection of cervical cancer and the monitoring of disease progression.

Various types of cell death patterns were found to be closely related to cancer eradication, such as apoptosis, necroptosis, pyroptosis, and ferroptosis [10,11,12]. In recent years, a novel cell death manner named cuproptosis, which is different from apoptosis, necroptosis, pyroptosis, and ferroptosis, was discovered. This novel cell death manner is called “cuproptosis” [13]. Cuproptosis was found to play an important role in the occurrence and development of various cancers, including melanoma, hepatocellular carcinoma, renal cell carcinoma, glioma, and lung adenocarcinoma [14,15,16,17]. Previous studies have also found that cervical cancer cells can reprogram cellular metabolic pathways so that these cervical cancer cells can benefit from increased expression of hexokinase 2, providing metabolites and intermediates of tricarboxylic acids for their growth [18]. Copper-induced cell death is closely related to lipid acylation of proteins, which is concentrated in the tricarboxylic acid cycle and is used for enzyme functions [13].

In this study, we investigated the relationship between the expression of cuproptosis-related lncRNAs and the stage and prognosis of cervical cancer. Based on the lncRNAs associated with cuproptosis, a new index named cuproptosis-related lncRNAs risk score (CRLs risk score) was proposed to evaluate the disease risk and survival status. We also analyzed the gene mutation pattern, immune-related components, and sensitivity to chemotherapy drugs of cervical cancer patients in different risk groups, and the results showed that there were significant differences in those aspects between the two different risk groups. Our predictive typing model has great potential in guiding the typing and treatment of cervical cancer in patients with CRLs risk score sensitivity and has proved its clinical value and significance.

## 2. Materials and Methods

### 2.1. Public Data Acquisition and Processing

The gene expression data, phenotype data, and corresponding survival information of 304 patients with cervical squamous cell carcinoma and endocervical adenocarcinoma (CESC) were obtained from the Cancer Genome Atlas (TGCA) database. The mutation data of TCGA-CESC cohort were downloaded from TCGA. The annotated lncRNAs were obtained from Gene Transfer Format file. A total of 19 cuproptosis-related genes (CRGs), 48 immune checkpoint genes (ICGs), and 7 chemoradiotherapy sensitivity-related genes (CRSGs) were collected [19]. DESeq2 was used to filter the differentially expressed genes (DEGs) with the following criteria baseMean ≥ 50, Log2FoldChange ≥ 0.5, and p.adj < 0.05.

### 2.2. Development of CRLs Risk Score

Pearson correlation analysis was carried out for the cuproptosis-related genes and lncRNAs with the following criteria |r| > 0.3. The patients with CESC were divided into low- and high-expression groups according to the median expression value. Univariate and multivariate Cox regression analyses were used to determine the OS independently related FI-DELs. According to previous reports [20,21], the CRLs risk score was constructed as following: $Risk\;socre=\sum_{i=1}^{n}Coef\left(Xi\right)*Exp$. Coef (Xi) is the regression coefficient of the CRLs, and Exp (Xi) represents the expression levels of CRLs. The Yonden index from training group divided CESC patients into high- and low-risk groups.

### 2.3. Immune Profile Analysis and Enrichment Analysis

The immune score and stromal score of each sample in TCGA-CESC cohort were calculated by the “estimate” package in R. The proportion of 28 immune cells and factors in the tumor microenvironment of each sample in TCGA-CESC cohort was evaluated via the CIBERSORT algorithm in R software. The infiltration data of immune cells and factors were downloaded from Tumor Immune Estimation Resource.

Gene Set Variation Analysis (GSVA), a GSE method that estimates variation in pathway activity over a sample population in an unsupervised manner, was used to carry out Kyoto Encyclopedia of Genes and Genomes (KEGGs) enrichment analysis [22].

### 2.4. Drug Therapy Response and Tumor Mutation Burden

The “oncoPrediction” R package was used to evaluate the drug susceptibility scores of common drugs in cervical cancer treatment plans in the TCGA cohort using the different expression genes.

After downloading the somatic mutation data from the TCGA website, we examined and integrated TCGA data using the “maftools” package and analyzed the differences in tumor mutation burden (TMB).

### 2.5. Statistical Analyses

An unpaired two-tailed Student’s *t*-test was used, as indicated. All results were expressed as mean ± SEM.

## 3. Results

### 3.1. Landscape of Genetic and Transcriptional Variations in CRGs in CESC

We summarized the mutations for the CESC patients with different vital statuses, including variation classification, variation type, and single nucleotide variation class. The top 10 mutations in surviving CESC patients were TTN, MUC4, PIK3CA, MUC16, KMT2C, RYR2, KMT2D, FLG, DMD, and FBXW7 (Figure 1a). The top 10 mutations in dead CESC patients were TTN, PIK3CA, KMT2D, MUC16, KMT2C, SYNE1, FLG, EP300, HUWE1, and LRP1B (Figure 1b). NFE2L2 was the gene with the highest mutation frequency (5%) among the 19 CRGs (Figure 1c). NFE2L2 was the gene with the highest mutation frequency (5%), followed by NLRP3 and PDHA1 (3%) (Figure 1d). Mutations in the gene differed between living and dead patients, but there was no significant difference in the TMB or TMB-related survival. (Figure 1e,f). The 19 CRGs were good at distinguishing CESC patients as measured by consensus analysis (Figure 1g), but there was no significant difference in survival between the two groups of CESC patients (Figure 1h). To determine their association with survival, we performed a univariate Cox regression analysis for these 19 CRGs and found none of the 19 CRGs was significantly associated with CESC survival (Figure 1i).

### 3.2. Screening of Biomarkers Associated with Cuproptosis in CESC

LncRNAs are major regulatory factors that regulate gene expression, play important roles in various biological processes, and can also be used as biomarkers for many cancers. Pearson correlation analysis was used to screen cuproptosis-related lncRNAs. A total of 1497 cuproptosis-related lncRNAs (CRLs) was identified. Through consensus cluster analysis, the CESC patients could well be distinguished (Figure 2a). CESC patients in CRLs cluster2 displayed worse OS (Figure 2b). Principal component analysis (PCA) showed CESC patients in cluster1 were well distinguished from cluster2. Machine learning analysis indicated that the accuracy of the 1497 CRLs in the prediction of survival situations was poor. Therefore, we randomized the patients into training and validation groups at 2:1 and performed univariate Cox regression analysis followed by least absolute shrinkage and selection operator analysis and multivariate Cox regression analysis (Figure 2e,f, Appendix A). Nine CRLs were determined to be associated with the overall survival (OS) of CESC. Patients with CESC with a high expression of AC002128.2 and AC009237.14 displayed worse OS, and those with a high expression of AC002563.1, AC048337.1, AC145423.1, AL117336.1, AP001542.3, ATP2A1-AS1, and LINC00426 displayed better OS (Figure 2g–o).

### 3.3. Construction and Validation of Risk Model in CESC

After candidate biomarkers determined by univariate and multivariate Cox regression analyses, those nine CRLs (AC002128.2, AC002563.1, AC009237.14, AC048337.1, AC145423.1, AL117336.1, AP001542.3, ATP2A1-AS1, and LINC00426) were used to construct a risk model according to previous studies. In groups of training, validation, and entire, we found that CESC patients with low-risk scores displayed longer survival times (Figure 3a,e,i). The Kaplan–Meier (K–M) survival curve showed that patients with low-risk scores had a better OS than those high-risk patients in the training, validation, and entire groups (Figure 3b,f,j). PCA showed that CESC patients with high-risk scores were well distinguished from CESC patients with low-risk scores using those nine CRLs (Figure 3c,g,k). The 1-year area under curve (AUC) values of risk models in the training, validation, and entire groups were all higher than 0.7, at 0.713, 0.737, and 0.716, respectively (Figure 3d,h,l). The AUC values of time-dependent receiver operating characteristic (ROC) curves of this model were all good. In the entire group, the AUC values of 1-year, 3-year, and 5-year ROC curves were all higher than 0.7 and were 0.716, 0.718, and 0.719, respectively.

To clarify the relationship for the risk score with pathologic M and pathologic N, we first carried out a difference analysis and found that there was no significant difference between CESC patients with M0 and M1 and CESC patients with N0 and N1. In M0 CESC patients, high-risk CESC patients still showed significantly poorer survival. In M1 CESC patients, high-risk CESC patients did not have significantly poorer survival (Figure 4b). The difference in overall survival between high- and low-risk patients was independent of N (Figure 4d). Univariate and multivariate Cox regression analyses were performed for the risk model and pathologic NM. The risk model was determined to be independent for the OS of CESC (Figure 4e,f). Compared with the pathologic NM, the risk model shows the best predictability (Figure 4g).

Next, we conducted a validation study. Consensus analysis showed those nine CRLs could better distinguish CESC patients (Figure 4h). There were also significant differences in survival between the two groups (Figure 4i). Patients with CESC can be basically separated using these nine biomarkers (Figure 4j). In terms of prognostic accuracy, the risk models we constructed all had higher AUC values than the different models of machine learning (Figure 4k).

### 3.4. Correlations of CRLs Risk Score with Immunotherapy, Chemoradiotherapy Sensitivity-Related Genes, and Immune Features in CESC

Then, we evaluated the tumor microenvironment (TME) and tumor-related score (TRS) status for the CESC patients in different risk groups. The ESTIMATE and immune were significantly increased, whereas the tumor purity significantly decreased in CESC patients with low risk scores (Figure 5a). The stemness score was significantly decreased in CESC patients with low risk scores (Figure 5b). The immune scores of 18 out of 28 immune cells and factors were significantly different between the high- and the low-risk groups, such as activated dendritic cell, CD56 bright natural killer cell, central memory CD4 T cell, effector memory CD8 T cell, eosinophil, and γ delta T cell (Figure 5c). The correlation of those nine CRLs and risk scores with the TME, TRS, and immune cells and factors were displayed in Appendix A. We also analyzed the immune infiltration between high- and low-risk patients, and the molecules with significant differences in infiltration are shown in Figure 5d. Of the 48 ICGs, 32 (30 significantly decreased and 2 significantly increased in the high-risk group) showed significant differences in gene expression between the high- and low-risk groups (Figure 5e). The results of the GSEA of KEGG showed that 18 signaling pathways were significantly enriched. Most of them were immune-related pathways, such as Th1 and Th2 cell differentiation, Th17 cell differentiation, T cell receptor signaling pathway, cytokine–cytokine receptor interaction, natural killer cell-mediated cytotoxicity, and PD-L1 expression and PD-1 checkpoint pathway in cancer (Figure 5f).

### 3.5. Correlations of CRLs Risk Score with CRSGs and Chemotherapeutic Sensitivity in CESC

To determine whether the CRLs risk model can guide the clinical use of chemical drugs, we firstly investigated the expression levels of nine CRSGs in different risk groups. EGFR and MGMT were significantly increased and decreased in the high-risk group, respectively (Appendix A). Then, we performed correlation analysis for those nine CRLs and risk scores with seven CRSGs and found some signatures do have significant correlations (Figure 6b,c).

OncoPredict algorithm was used to evaluate the sensitivity of chemotherapeutic drugs. The sensitivity of 18 chemotherapeutic drugs varied significantly between the high- and low-risk groups (Figure 6d). Among them, there were 7 drugs with significantly increased sensitivity to chemotherapy drugs in the high-risk group, and 11 drugs with significantly decreased sensitivity to chemotherapy drugs in the high-risk group. In addition, correlation studies found that three drugs (CDK9_5038_1709, Dihydrorotenone_1827, and Vincristine_1818) were significantly positively correlated with risk score, and ten drugs (Docetaxel_1007, Eg5_9814_1712, Staurosporine_1034, Dinaciclib_1180, Docetaxel_1819, Bortezomib_1191, MG-132_1862, Dactolisib_1057, Rapamycin_1084, and Luminespib_1559) were significantly negatively correlated with risk score. Three drugs, including Bortezomib_1191, Luminespib_1559, and Rapamycin_1084, were also associated with risk values in the high-risk group (Appendix A).

## 4. Discussion

Cervical cancer remains the fourth most common cancer among women. Early cervical cancer has a high 5-year overall survival (OS) rate of 90% thanks to early screening and effective surgery, chemotherapy, and radiotherapy [7]. But metastatic cervical cancer has a poor survival rate, with a 5-year survival rate of less than 10% [7]. Therefore, screening of cervical cancer biomarkers is of great significance for its early diagnosis and progress monitoring. Copper is a co-factor of many enzymes, but it also leads to a series of cell metabolic disorders and eventually cell death due to its accumulation. Cuproptosis is a new type of cell death, which is different from apoptosis, necrosis, pyroptosis, and ferroptosis, caused by the accumulation of lipoylated dihydrolipoamide S-acetyltransferase, induced by excessive copper in cells, and then leads to protein toxic stress [23]. Previous studies have found that serum levels of copper ions increase significantly in patients with various cancers, such as lung cancer, prostate cancer, breast cancer, gallbladder cancer, gastric cancer, and thyroid cancer [24,25,26,27,28,29]. Moreover, previous studies also found that the lower the clinical staging of lung cancer patients, the higher the serum copper ion concentration, and the higher the serum copper ion concentration, the worse the clinical prognosis [30,31]. In addition, activation of many cancer-related signaling pathways is affected by copper ions, including the phosphoinositide-3-kinase (PI3K)-AKT signaling pathway [32], FoxO signaling pathway [33], and mitogen-activated protein kinase (MAPK) signaling pathway [34]. Many cuproptosis-related signatures can be used as biomarkers of several cancers, including renal cell carcinoma, breast cancer, hepatocellular cancer, and endometrial cancer [15,35,36,37].

Through a series of analyses, we found that nine CRLs (AC002128.2, AC002563.1, AC009237.14, AC048337.1, AC145423.1, AL117336.1, AP001542.3, ATP2A1-AS1, and LINC00426) could be used as prognostic biomarkers for CESC patients, and the constructed risk model using those nine CRLs could better predict the prognosis of CESC. Previous studies also found that these markers can indeed be used as markers of cancer. For example, Zhou et al. found that AC002128.2 could be the prognostic biomarker for lung adenocarcinoma [38]. Zhang, Liang, Chen, et al. found that AC009237.14 could be the prognostic biomarker for colon adenocarcinoma [39,40,41]. Zeng et al. found that ATP2A1-AS1 could be a prognostic biomarker for patients with myeloma [42]. Feng et al. found that ATP2A1-AS1 could be the prognostic biomarker for cervical cancer [43]. LINC00426 could be the prognostic biomarker for breast cancer [44], lung cancer [45,46], and renal cancer [47]. These previous findings suggest that those lncRNAs may serve as prognostic biomarkers for a variety of cancers. In our study, we found that these lncRNAs may also serve as prognostic markers of CESC. Our results further enrich the possibility of these lncRNAs as biomarkers for cancers. In predicting CESC prognosis, the AUC value of this risk model was relatively high, at 0.727. Compared with previous cervical cancer biomarkers, although the AUC value of our risk model is not the highest, similar to many previous risk models, our risk model can not only predict prognosis better but also serves as an independent prognostic model [48,49,50,51]. These results also suggest that we may need to further refine our models. Previous studies found that risk models also play an important role in guiding clinical drug use in cancer patients. For example, Bing et al. found that CDC20 and ARID4A were significantly associated with good and poor prognoses following chemotherapy [52]. Weng et al. found that SCG2 was a risk factor with higher expression predicting poorer prognosis, and the SCG2 high expression subgroup was the immune hot type and considered more suitable for immunotherapy [53]. Weng et al. found that a generated metabolic risk score model using a 21-gene mRNA dryness index could provide a more accurate stratification of colorectal risk and screening of colorectal patients who respond to immunotherapy [54]. In our present study, we found that this risk model was also closely related to several ICGs, CRSGs, and immune cells and factors, and many chemotherapy agents differ significantly between high- and low-risk CESC patients.

CESC is characterized by some alterations in the immune system, and it is thought to be a typical immunogenic cancer caused by HPV infection [55]. In the present study, we did find significant changes in the immune microenvironment in CESC patients (including several immune cells and factors), which further proves that the immune system is closely related to the occurrence of CESC [56,57]. The analysis of KEGG enrichment analysis indicated that several immune-related signaling pathways were enriched, including Th1 and Th2 cell differentiation, Th17 cell differentiation, T cell receptor signaling pathway, cytokine–cytokine receptor interaction, PD-L1 expression, and PD-1 checkpoint pathway in cancer. These results also further suggest that immunotherapy based on the regulation of immune-related signaling pathways may be another very effective option for CESC patients [58,59,60].

In addition, we found significant differences and correlations between high- and low-risk patients with high sensitivity to drugs, such as Bortezomib_1191, Luminespib_1559, and Rapamycin_1084, suggesting that these three drugs may be used in the treatment of high-risk CESC patients. Previous studies found that Bortezomib was indeed effective in the treatment of CESC patients [61,62,63,64]. In combination therapy, Bortezomib significantly increases the sensitivity of multiple agents to CESC [61,62,63,64]. The mechanistic target of Rapamycin (mTOR) is an atypical serine/threonine kinase that plays an important role for several cancers, including cervical cancer [65,66,67]. Regulating the activation of the mTOR signaling pathway can effectively save the proliferation, migration, and invasion for CESC [65,66,67]. The treatment of CESC by Rapamycin was similar to that of Bortezomib for CESC. Luminespib has been found to be useful in treating a variety of cancers, such as lung cancer and gastric cancer [68,69]. In this study, we found that patients with CESC are sensitive to the drug Luminespib, which also expands the chemoradiotherapy strategy for CESC. Our study also found that Bortezomib and Rapamycin were sensitive to CESC patients, which further confirmed the role of Bortezomib and Rapamycin in CESC. Our results are consistent with previous reports, which further indicates that our research strategy is feasible and our results are credible.

## 5. Conclusions

A comprehensive analysis showed that nine CRLs (C002128.2, AC002563.1, AC009237.14, AC048337.1, AC145423.1, AL117336.1, AP001542.3, ATP2A1-AS1, and LINC00426) were independently associated with OS in CESC patients. Risk models constructed by these nine CRLs can not only well predict the prognosis of CESC, but also have significant correlation with immunotherapy- and chemotherapeutic-related genes, and can also well predict the sensitivity of several chemotherapy drugs, providing guidance for clinical medication based on this prognostic model. However, further validation studies are needed.

## Figures and Tables

**Figure 1 genes-14-01381-f001:**
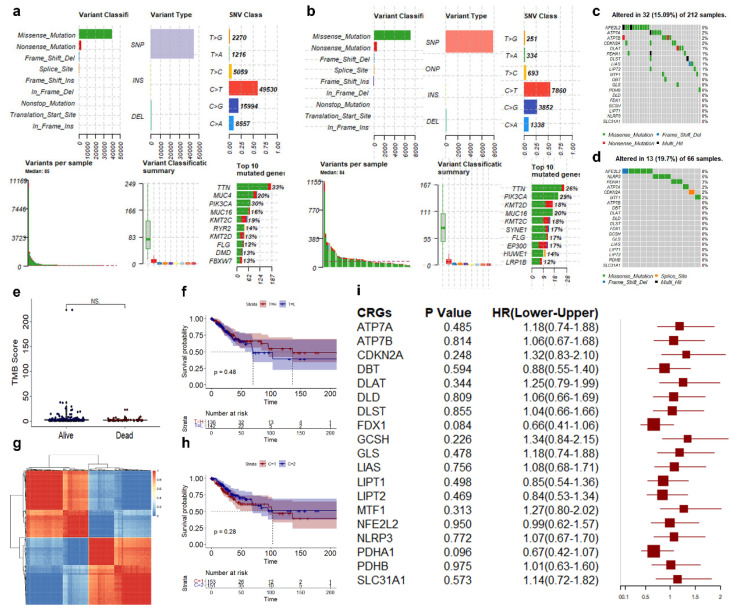
Landscape of genetic and transcriptional variations in CRGs in CESC. (**a**,**b**) Summary of variation in surviving CESC patients (**a**) and dead CESC patients (**b**). (c,**d**) Genetic variations landscape of 19 CRGs in surviving CESC patients (**c**) and dead CESC patients (**d**). (**e**) Differential analysis of TMB. (**f**) K–M curve of TMB score. (**g**) Result of cluster analysis using 19 CRGs. (**h**) K–M curve of different cluster using 19 CRLs. (**i**) Univariate Cox regression result of 19 CRGs. NS: no significance.

**Figure 2 genes-14-01381-f002:**
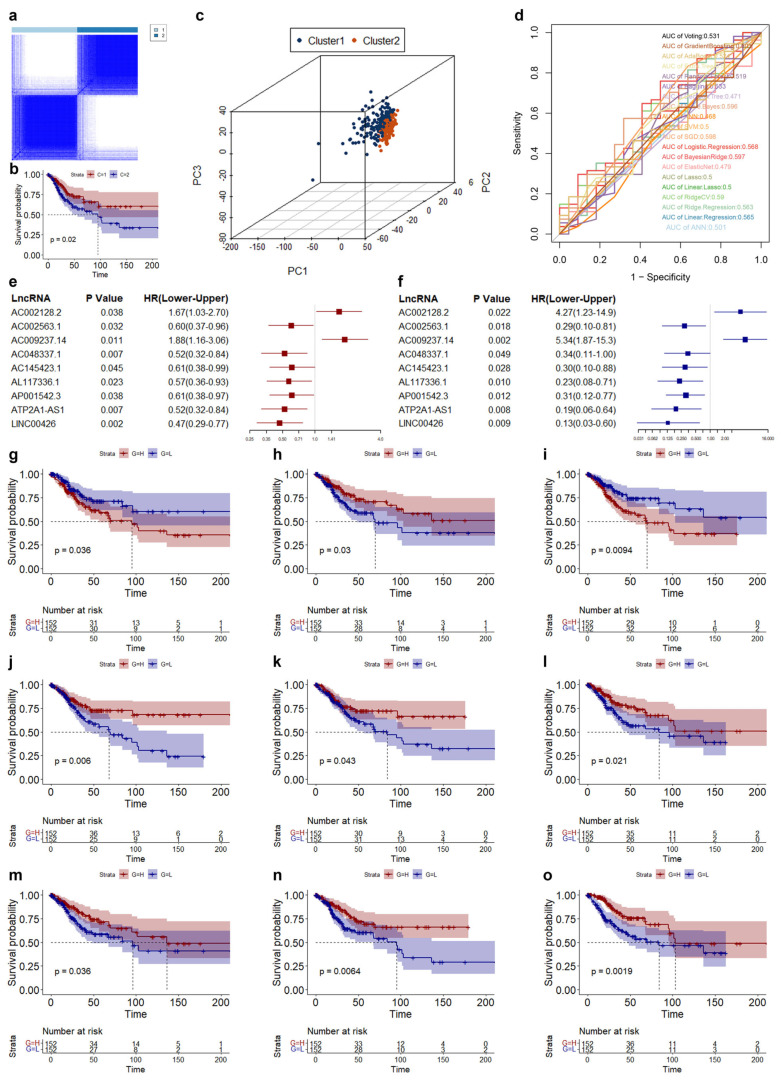
Screening of OS-related biomarkers in CESC. (**a**) Result of cluster analysis using 1497 CRLs. (**b**) K–M curve of different cluster using 1497 CRLs. (**c**) Distribution of CESC in different cluster. (**d**) ROC curve of various models using 1497 CRLs. (**e**,**f**) Results of univariate and multivariate Cox regression. (**g**–**o**) K–M curve of AC002128.2 (**g**), AC002563.1 (**h**), AC009237.14 (**i**), AC048337.1 (**j**), AC145423.1 (**k**), AL117336.1 (**l**), AP001542.3 (**m**), ATP2A1_AS1 (**n**), and LINC00426 (**o**).

**Figure 3 genes-14-01381-f003:**
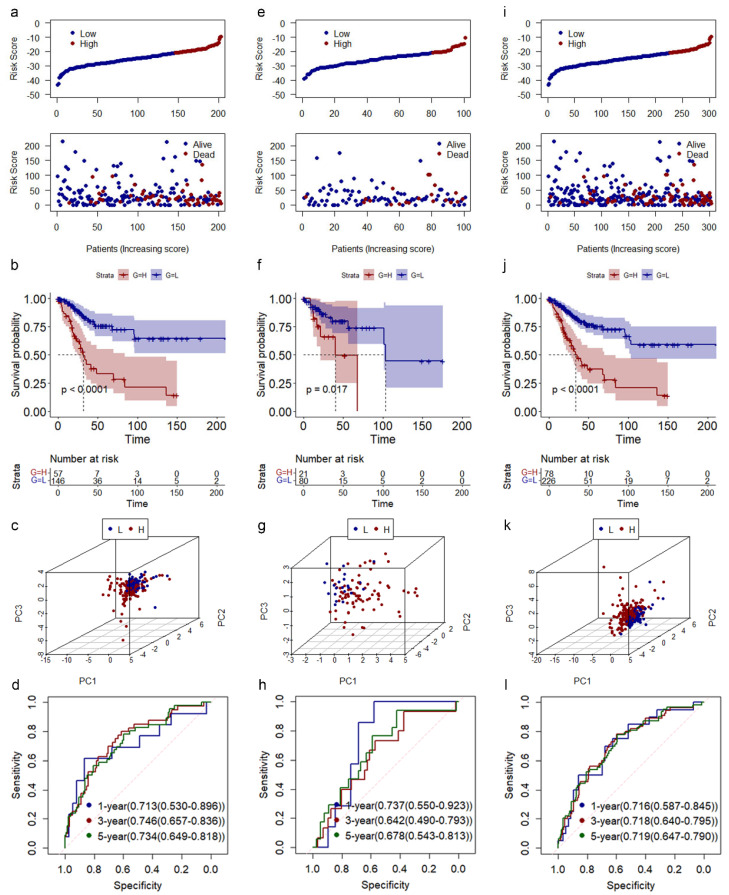
Construction and validation of risk model in CESC. (**a**–**d**) Risk score distribution (a up), survival time (a down), K–M curve (**b**), PCA results (**c**), and time-dependent ROC curve (**d**) in training group. (**e**–**h**) Risk score distribution (e up), survival time (e down), K–M curve (**f**), PCA results (**g**), and time-dependent ROC curve (**h**) in training group. (**i**–**l**) Risk score distribution (i up), survival time (i down), K–M curve (**j**), PCA results (**k**), and time-dependent ROC curve (**l**) in training group.

**Figure 4 genes-14-01381-f004:**
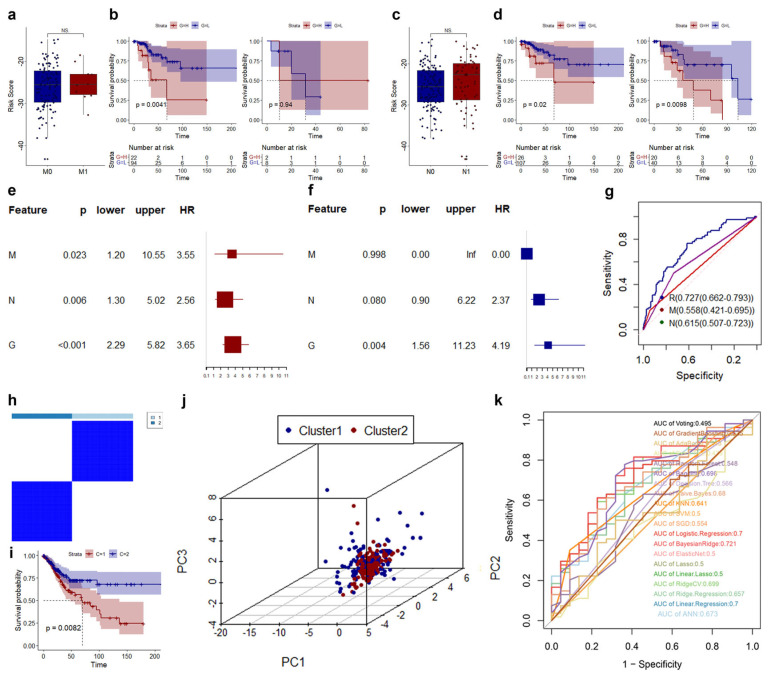
Evaluation of the CRLs prognostic model and different clinical features. (**a**) Correlation of risk score and pathologic M. (**b**) K–M curve of risk model in M0 (left) and M1 (right). (**c**) Correlation of risk score and pathologic N. (**d**) K–M curve of risk model in N0 (left) and N1 (right). (**e**,**f**) Univariate and multivariate Cox regression analyses for risk model and pathologic NM. (**g**) ROC curve of risk model and pathologic NM. (**h**) Result of cluster analysis using 9 CRLs. (**i**) K–M curve of different clusters using 9 CRLs. (**j**) Distribution of CESC in different clusters. (**k**), ROC curve of various models using 9 CRLs.

**Figure 5 genes-14-01381-f005:**
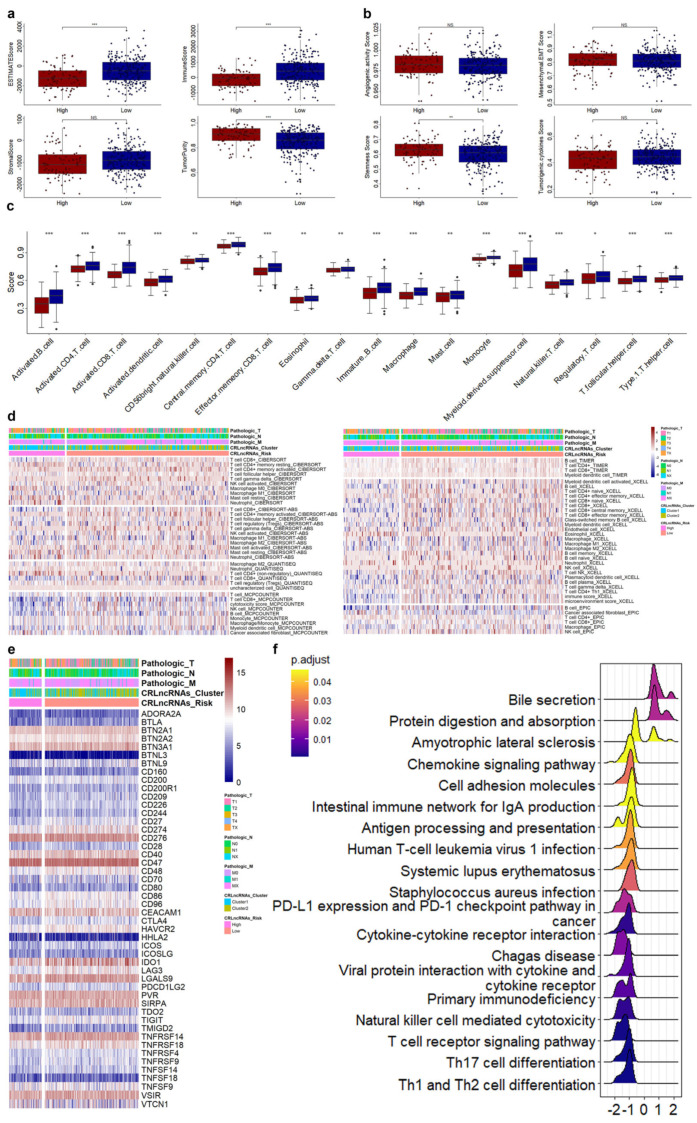
Evaluation of immunity in CESC patients with different risk scores. (**a**,**b**) Expression of tumor microenvironment (**a**) and tumor-related scores (**b**) in high- and low-risk groups. (**c**) Immune cells and molecules with significant differences in immune scores. (**d**) Immune cells and molecules with significant differences in immune infiltration. (**e**) Expression of ICGs between high- and low-risk groups. (**f**), Signaling pathways enriched by differentially expressed genes between high- and low-risk groups. *, *p* < 0.05. **, *p* < 0.01. ***, *p* < 0.001.

**Figure 6 genes-14-01381-f006:**
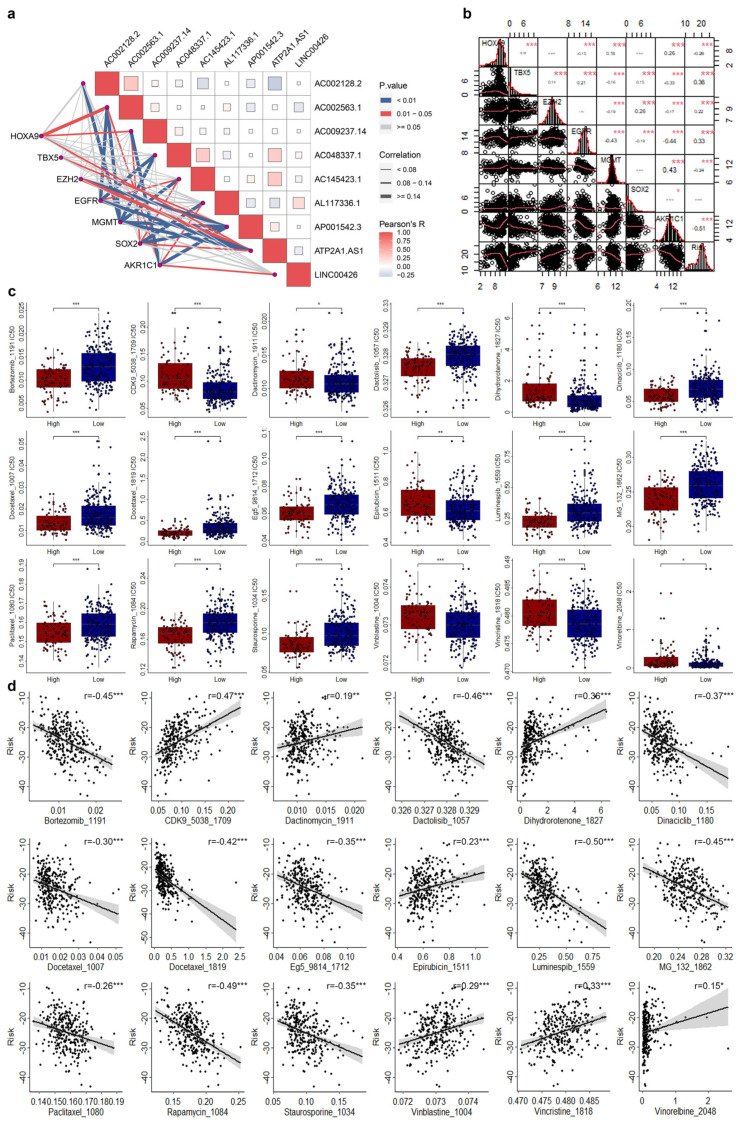
Correlations of risk scores with CRSGs and chemotherapeutic drug sensitivity in CESC. (**a**) Correlation of seven CRSGs and nine CRLs. (**b**) Correlation of seven CRSGs and risk scores. (**c**) Difference analysis for the drug sensitivity in high- and low-risk groups. (**d**) Correlation of drug sensitivity and risk score. *, *p* < 0.05. **, *p* < 0.01. ***, *p* < 0.001.

## Data Availability

The data that support the findings of this study are openly available in TCGA at https://portal.gdc.cancer.gov/ (accessed on 11 January 2023).

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
