# Peer review of "Prediction of Prognosis and Chemotherapeutic Sensitivity Based on Cuproptosis-Associated lncRNAs in Cervical Squamous Cell Carcinoma and Endocervical Adenocarcinoma"

_genes, 2023, doi:10.3390/genes14071381_

Round 1
Reviewer 1 Report
The article “Prediction of prognosis and chemotherapeutic sensitivity based 2 on Cuproptosis-associated lncRNAs in cervical squamous cell 3 carcinoma and endocervical adenocarcinoma” is a comprehensive analysis showing that nine CRLS were independently associated with OS in CESC patients.
Cervical cancer remains the fourth most common cancer among women. Early cervical cancer has a high 5-year overall survival rate (OS) of 90% thanks to early screening and effective surgery, chemotherapy, and radiotherapy. But metastatic cervical cancer has a poor survival rate, with a 5-year survival rate of less than 10 percent. Therefore, screening cervical cancer biomarkers is of great significance for its early diagnosis and progress monitoring. Copper is a cofactor of many enzymes, but it also leads to a series of cell metabolic disorders and eventually cell death due to its accumulation. Cuproptosis is a new type of cell death, which is different from apoptosis, necrosis, pyroptosis, and ferroptosis, which is caused by the accumulation of lipoylated dihydrolipoamide S-acetyltransferase, induced by excessive copper in cells, and then leads to protein toxic stress. Many cuproptosis-related signatures can be used as biomarkers of several cancers, including renal cell carcinoma, breast cancer, hepatocellular cancer, and uterine oorpus endometrial cancer.
The authors found significant differences and correlations between high- and low-risk patients with high sensitivity to drugs Bortezomib, Luminespib, and Rapamycin, suggesting that these three drugs may be used in the treatment of high-risk CESC patients. In this study, the authors found that patients with CESC are sensitive to drug A, which also expands the chemoradiotherapy strategy for CESC.
Overall, the manuscript is written well, and reasonable conclusions were drawn.
Author Response
Point 1: Overall, the manuscript is written well, and reasonable conclusions were drawn.
Response 1: Thank you for you positive comments.
Reviewer 2 Report
To The Chief Editor
Genes
The manuscript “Prediction of prognosis and chemotherapeutic sensitivity based on Cuproptosis- associated lncRNA in cervical squamous cell carcinoma and endocervical adenocarcinoma” is well written and a very good attempt to show an association between gene mutation, immune-related component and sensitivity of chemotherapeutics drug to cervical patient carcinoma through bioinformatic analysis. All experiments are well planned, and the results validate the objective of the study. Before this paper can be accepted for publication some points need to be addressed.
Major issues.
1. Author should improve the quality of the image uploaded in Figures 1(a, b, c, and d), 2 (e and f), 5 (d and e), and 6 (b and c) it is very hard to analyze the data.
2. Did the author compare the expression pattern of nine deregulated lncRNA in the tissue of cervical cancer patients vs the normal tissue other than using a bioinformatics tool?
3. Author should discuss the results of the immune-related pathway and how it is correlated with the prognosis of cervical cancer patients.
4. The result obtained from this study can only be used for the prediction of the prognosis of cervical cancer, to validate its results experiment is needed to be performed either by in vitro or invivo models.

The quality of the english used in the paper can be improved.
Author Response
Point 1. Author should improve the quality of the image uploaded in Figures 1(a, b, c, and d), 2 (e and f), 5 (d and e), and 6 (b and c) it is very hard to analyze the data.
Response 1: Thank you for you great suggestions. We have revised it according to your suggestions.
Point 2. Did the author compare the expression pattern of nine deregulated lncRNA in the tissue of cervical cancer patient’s vs the normal tissue other than using a bioinformatics tool?
Response 2: Thank you for you great suggestions. It is very important to verify the expression of target genes in clinical samples. Because we have not obtained enough CESC samples, we cannot carry out this part of the study at present. This is also the research we will carry out in the future.
Point 3. Author should discuss the results of the immune-related pathway and how it is correlated with the prognosis of cervical cancer patients.
Response 3: Thank you for you great suggestions. We have added the detail discussion about the immune-related pathway according to your suggestions in the revised manuscript.
Point 4. The result obtained from this study can only be used for the prediction of the prognosis of cervical cancer, to validate its results experiment is needed to be performed either by in vitro or in vivo models.
Response 4: Thank you for your great suggestions. It is very important and necessary to obtain clinical samples to carry out validation studies on prognostic risk models and functional studies on candidate biomarkers signatures. However, we have not obtained enough clinical samples for CESC at present, and our current research is mainly focus on the bioinformatics. The above two studies are also the main research contents of our later plan.